# Prevalence and correlates of multimorbidity among adults in Botswana: A cross-sectional study

**Mpho Keetile** *, **Kannan Navaneetham, Gobopamang Letamo**

Department of Population Studies, University of Botswana, Gaborone, Botswana

* mphokeet@yahoo.com

**Data Availability Statement:** The research permit and ethical approval was obtained from the Ministry of Health and Wellness, Government of Botswana. Data cannot be shared without the approval of Government of Botswana. Data can be

## Abstract

### Background

Botswana is currently undergoing rapid epidemiological transition indicated by a decline in infectious diseases and an increase in chronic non-communicable diseases and their associated risk factors. The main aim of this study was to assess prevalence and correlates of multimorbidity among the adult population in Botswana.

### Methods

A cross-sectional study called Chronic Non-Communicable Diseases Study (NCDs study) was conducted in March, 2016. Using multistage cross sectional sampling design, 1178 male and female respondents aged 15 years and above were interviewed across 3 cities and towns, 15 urban villages and 15 rural villages. Participants were interviewed face-to-face using a structured questionnaire. Adjusted multinomial logistic regression analysis was used to assess covariates of multimorbidity. The statistical significant level was fixed at p <0 .05.

### Results

Prevalence of multimorbidity in the sampled population was estimated at 5.4%. Multivariate results indicate that the odds of multimobridty were significantly high among women (AOR = 3.34, 95% C.I. = 1.22–21.3) than men. On the other hand, the odds of multimorbidity were significantly low among young people aged below 24 years (AOR = 0.01, 95% C.I. = 0.00–0.07), currently married people (AOR = 0.24, 95% C.I. = 0.07–0.80) and individuals in the 2nd wealth quintile (AOR = 0.20, 95% C.I. = 0.05–0.75) compared to their counterparts. For behavioural risk factors, alcohol consumption (AOR = 4.80, 95% C.I. = 1.16–19.8) and overweight/obesity (AOR = 1.44, 95% CI = 1.12–2.61) were significantly associated with high multimorbidity prevalence.

### Conclusion

Multimorbidity was found to be more prevalent among women, alcohol consumers and overweight/obese people. There is need to strengthen interventions encouraging healthy

available to other researchers with the approval by the Government of Botswana. Data access requests may be sent to Dr. Serai Daniel Rakgoasi, Department of Population Studies, University of Botswana, email;RAKGOASI@mopipi.ub.bw.

**Funding:** The NCDs survey was supported by Office of Research and Development, through Round 28 Research Funding, University of Botswana.However, the Office of Research and Development has no role in the analysis, decision to publish or preparation of the manuscript.

**Competing interests:** The authors have declared that no competing interests exist.

lifestyles such as non-consumption of alcohol, physical activity and healthy diets. Moreover, there is need for a holistic approach of health care services to meet the needs of those suffering from multimorbidity.

## Introduction

The burden of non-communicable diseases (NCDs) is increasing globally [1] due to demographic and health transitions. It has been estimated that globally NCDs are the leading cause of death and they account for 68% per cent (38 million) of all deaths [2]. Of all the deaths due to NCDs it was estimated that almost three quarters of them (28 million) occur in LMICs [3]. Consequently, it was also projected that the deaths from infectious diseases would decline and the NCD deaths would increase in the future [1] especially in developing countries. These conditions would reduce quality of life of the population as well as cause substantial financial burden to governments and the society as a whole [4]. In recent years, the proportion of people who have been diagnosed to have multimorbidity (two or more chronic conditions) is increasing and is likely to pose serious health threats and financial burden on the health systems and populations [5].

Before the 1980s, common diseases in Botswana were infectious diseases and those associated with unsanitary conditions, poverty and inadequate hygiene [6]. The increase of NCDs in Botswana, as in many LMICs can be attributed to urbanization and the changing lifestyles as well as the improved standard of living. These include improved road infrastructure and increased volume of traffic, as well as high levels of food and alcohol consumption [7]. Since the 1980s new patterns of conditions associated with affluent lifestyles such as hypertension, diabetes and cardiovascular diseases emerged [6]. The magnitude of such diseases was overshadowed by the re-emergence of infectious diseases such as tuberculosis and HIV/AIDS [8].

There has been an increase in the prevalence of NCDs and their risk factors in the past few decades. Common risk factors for NCDs include; tobacco use, unhealthy diet, poor physical activity and excessive use of alcohol [7]. NCDs such as heart disease, stroke, cancer and other chronic diseases and associated risk factors are now becoming the main causes of morbidity and mortality in the population [8]. These result from unhealthy diets, low physical activity, exposure to tobacco smoke or the effects of the harmful use of alcohol [2]. Furthermore, the key drivers of NCDs are aging, rapid unplanned urbanisation and unhealthy lifestyles [1]. Unhealthy lifestyles like unhealthy diets, for instance, may show up in individuals as raised blood pressure, increased blood glucose, elevated blood lipids, and overweight or obesity [2].

Surveillance of NCDs and NCD risk factors has been on-going in recent times in Botswana. The first surveillance survey was conducted in 2007, and it indicated that the major risk factors related to NCD were at a higher rate. The second survey was conducted in 2014, to explore and compare the results to see if at all the interventions taken has brought change in the disease burden. Both surveys have indicated that both NCDs and NCD risk factors are high in the adult population. For instance, in both 2007 and 2014 surveys prevalence rates of NCDs risk factors such as smoking (19.7% vs 18%), unhealthy diet (96.6%vs 94.8%), poor physical activity (34.7% vs 20.1), alcohol consumption (54.1% vs 26%) and overweight/obesity (38.6% vs 30.6%) were high. Moreover, prevalence of hypertension (33.3% in 2007 vs 35.2% in 2014) and diabetes (2.3% in 2007 vs 2.7% in 2014) had increased [8,9].

Given that NCDs are now common in Botswana, it is important to examine if there is NCDs clustering and to identify which SES groups are most affected. It has been observed that

NCD clustering has great variation in its distribution among different socioeconomic groups within societies [2]. There is paucity of evidence on studies assessing NCD clustering in Botswana. This study is the first to assess patterns and correlates of NCD multimorbidity in Botswana. Consequently, more insight into patterns and correlates of multimorbidity is important for policy purposes and for providing an understanding of the factors that are likely to contribute to NCD clustering in Botswana. The main objective of this study is to estimate the prevalence of multimorbidity and to identify its correlates.

## Materials and methods

### Study design, setting and sampling

A cross-sectional secondary data from the survey on Chronic Non-Communicable Diseases in Botswana (NCDs survey) was used for analysis. The NCDs survey was conducted in March 2016. Multistage probability sampling design was used for the survey. At the first stage, census districts were divided into rural and urban clusters. At the second stage, urban districts were divided into cities or towns and urban villages while rural clusters were maintained. The third stage comprised a random selection of 3 cities and towns in the cities and towns strata, 15 urban villages from urban villages' strata and 15 rural villages' from the rural areas strata.

The final and fourth stage was the selection of enumeration areas using probability proportional to size sampling method for the different strata and localities. For each selected enumeration area (EA), 20 households were selected using systematic sampling method. This followed guidelines used in most demographic health surveys (DHS) where 20–25 households (HHS) are selected from the primary sampling units (PSUs) [8]. The Kish grid was used to select eligible respondents from the selected households. From an estimated initial sample size of 1280, 1178 respondents aged 15 years and over who had successfully completed the individual questionnaire were interviewed yielding a response rate of 92 per cent. The inclusion criteria for this study were such that all individuals who had successfully completed the NCD study questionnaire were considered for analysis, while those who did not were excluded from the sample.

### Data collection

After obtaining verbal informed consent, participants were interviewed face-to-face by trained research assistants with a minimum qualification of a bachelor's degree in the social sciences using structured paper questionnaire. The research assistants were trained on field survey methodology, interview skills and research ethics. These were mainly based on the WHO Training Manual on the Study on Global Ageing and Adult Health (SAGE) and other documents were used for reinforcement. The NCDs survey adopted the modified de facto type of enumeration whereby respondents above 15 years old were enumerated at the place where they were found at the time of survey using the interviewer (canvasser) method. In enumerating an EA, a coin was tossed to decide the cardinal point where the enumeration would start. The first household to be interviewed was determined using the day code. For example, on the 25th March 2016 –the first household to be enumerated would be the 7th household from the farthest point of the EA. This code was arrived at by adding the digits 2 and 5. At a sampled household a listing of all people 15 years and above was done. If there was more than one eligible participant in the selected household, one older person was selected to participate by a lottery method. If the eligible older person was absent during the first data collection visit, the interviewer arranged to return at another time to do the interview.

## Ethical considerations

All ethical clearance formalities were completed before the start of the study. The study proposal along with the necessary documents were submitted to and approved by the Institutional Review Board of the University of Botswana. Approval for conducting the study was also obtained from the Government of Botswana through the Ministry of Health and Wellness. Privacy and confidentiality of the highest standard were maintained by treating all respondents as anonymous and no names of respondents are mentioned or implied when presenting findings of the study.

## Measures

The outcome variable for this study is self-reported multimorbidity. This was a composite variable created to assess the clustering of NCD conditions among individuals in the study. From the list of NCD conditions, the study included the following 10 chronic conditions; Stroke, angina, diabetes, chronic lung disease, asthma, hypertension, eye/vision problem, nerves problem, skin problem and depression. The variable was created from NCD conditions reported in the study population and was coded such that if there was no existence of any NCD conditions, a code of 0 was given and if there was any one NCD condition, the code was given as 1 = single NCD condition and the code was given as 2 if there were more than one NCD conditions. Henceforth, multimorbidity is defined in this study as the presence of two or more chronic NCD conditions in an individual, an analogue to previous research [10,11].

NCD risk factors were used as explanatory variables. The NCD risk factors were derived using the WHO standard questionnaire and the variables were constructed as shown below;

**Tobacco use.**   Tobacco use was measured as the percentage of individuals who currently smoke any tobacco products such as cigarettes, cigars or pipes.

**Alcohol use.**   Alcohol consumption was measured based on the intensity of alcohol consumed in the past 30 days. Respondents who had consumed alcohol in the past 30 days were asked about the number of standard alcohol drinks they had each day in the past 7 days and if they reported to have had three or more drinks per day (of approximately 60 g alcoho1) it was considered to be excessive drinking and it was coded as 1 and 0 if otherwise.

**Poor consumption of fruit and/or vegetables.**   Consumption of fruits and vegetables was assessed in terms of 'number of servings'. Poor fruit and vegetables consumption was created when an individual reported daily consumption of less than the recommended 5 servings of fruit and vegetables consistent with WHO recommendation on diet [12]. Respondents reported the number of servings for fruits/vegetables they had in a typical day, and if the servings were less than 5 in a day, they were considered to be having poor fruit/vegetable consumption. The final variable was coded such that individuals with poor fruit and/or vegetable consumption = 1 and 0 if otherwise.

**Poor physical activity.**   Poor physical activity was calculated using an average of the typical types of activity undertaken [13]. It was calculated based on the time taken during physical activity in the past 7 days. Respondents were asked whether they do any moderate to rigorous intensity activities for at least 10 minutes continuously. This was considered for the domains of work and walking (includes at work and at home, walking to travel from place to place, and any other walking for recreation, sport, exercise, or leisure). Based on the time taken by respondents on work and walking, they were grouped into four categories of: no activity, low, moderate and high activity levels to show the intensity of their physical activity. The resultant variable was coded such that no and low activity (<10 mins of physical activity) = 1 (that is physically inactive) and moderate and high activity (≥10 mins of physical activity) were coded = 0 (that is physically active).

**Body mass index.** Body mass index (BMI) was derived from a ratio of weight in kilograms divided by height in meters squared. Height was measured in centimeters during the survey but was later converted to meters during the analysis. In this study, BMI was categorized into four groups as per WHO recommendations [1]: underweight (BMI < 18.5 kg/m$^2$), normal (18.5 kg/m$^2$ ≤ BMI < 25 kg/m$^2$), overweight (25 kg/m$^2$ ≤ BMI < 30 kg/m$^2$), and obese (BMI ≥ 30 kg/m$^2$). These categories were used to create a binary outcome variable which was coded as: being overweight and obese (BMI≥25) = 1; not overweight and obese = 0 (BMI<25).

## Socioeconomic and demographic variables

The other independent variables used for analysis in this study were selected on the basis of literature review. These variables include age, sex, marital status, work status, residence, wealth status and education.

Wealth status variable was created from the wealth index. Information on a range of durable assets was collected during the survey (e. g. ownership of car, refrigerator, and television,), housing characteristics (e. g. material of dwelling floor and roof, main cooking fuel), access to basic services (e. g. electricity supply, source of drinking water, sanitation facilities) and ownership of livestock (e.g. cattle, goats, sheep, horses, chickens). From this information principal component analysis was employed to derive the wealth index variable, which had five categories from the 1st to the 5th quintile (poorest to richest). For construction of other variables, standard categories have been used as directly derived from the data. The other socioeconomic variables were constructed as per conventional standard category.

## Statistical analysis

Descriptive analysis was done to assess the prevalence of multimorbidity, while multinomial logistic regression analysis was used to examine factors associated with multimorbidity in the population. For descriptive analyses frequencies and chi-square tests were used. Three models were run using multinomial logistic regression analysis. In model I we assessed the association between a single NCD condition and socioeconomic covariates. Model II presents results for the association between multimorbidity and socioeconomic variables. Model III- presents results for the association between multimorbidity and NCD risk factor variables, using socioeconomic variables as covariates. The reference group were individuals who did not report any NCD condition.

Adjusted odds ratios (AOR) together with their 95% confidence intervals (CI) were used for the interpretation of the results of multinomial logistic regression models. All comparisons were considered to be statistically significant at p <0.05 level. Since a multi-stage stratified sampling procedure was used in the NCDs survey, the use of standard statistical methods for analyzing the data would produce unreliable estimates of the desired parameters. As such during analysis of data, a complex sample module in SPSS was used to account for the hierarchical structure of the sampling design. Data analysis was done using SPSS version 25.

## Results

### Socio-demographic characteristics of the sample

Table 1 presents the socio demographic characteristics of the study population. There were a high proportion of females (69.1%) than males (30.9%). The sample age distribution indicates a relatively young population, with over half (59%) of the sample being less than 39 years of age, and almost three quarters (73.5%) being less than fifty years of age. A high proportion

**Table 1. Socioeconomic characteristics of the study population (N = 1178)-NCD survey, 2016.**

| Variable | Percentage (%) | Frequency (N) |
|---|---|---|
| **Sex** | | |
| Male | 30.9 | 364 |
| Female | 69.1 | 813 |
| **Age in years** | | |
| <24 | 26.4 | 270 |
| 25–34 | 29.5 | 302 |
| 35–44 | 19.2 | 196 |
| 45–54 | 12.7 | 130 |
| 55–64 | 7.3 | 75 |
| 65+ years | 4.9 | 50 |
| **Locality Type** | | |
| Cities/Towns | 30.2 | 355 |
| Urban Villages | 45.4 | 534 |
| Rural Settlements | 24.5 | 288 |
| **Marital Status** | | |
| Never Married | 73.8 | 864 |
| Currently married | 17 | 199 |
| Formerly married | 9.2 | 108 |
| **Highest Level of Education Attained** | | |
| Primary or Less | 35.5 | 410 |
| Junior Secondary | 27.2 | 314 |
| Senior Secondary | 17.3 | 200 |
| Tertiary & Over | 19.9 | 230 |
| **Work Status in past 12 months** | | |
| Public Sector | 10.5 | 122 |
| Private Sector | 15.7 | 182 |
| Self Employed | 11.2 | 130 |
| Not Employed | 37.5 | 436 |
| Homemaker-Student | 18.8 | 218 |
| Retired-Other | 6.4 | 74 |
| **Wealth status** | | |
| Lowest | 19.9 | 234 |
| Second | 20.1 | 237 |
| Middle | 19.9 | 235 |
| Fourth | 20.1 | 237 |
| Highest | 19.9 | 235 |
| **Smoking** | | |
| Yes | 11.6 | 137 |
| No | 88.4 | 1041 |
| **Alcohol consumption** | | |
| Yes | 17.3 | 204 |
| No | 82.7 | 974 |
| **Poor physical activity** | | |
| Yes | 48.9 | 576 |
| No | 51.2 | 602 |
| **Poor fruit and vegetable consumption** | | |
| Yes | 82.5 | 972 |

(*Continued*)

**Table 1.** (Continued)

| Variable | Percentage (%) | Frequency (N) |
|---|---|---|
| No | 17.5 | 206 |
| **Overweight/Obesity** | | |
| Yes | 41.3 | 487 |
| No | 58.7 | 691 |
| *Total* | | **1178** |

(45.4%) of the population resided in urban villages; while just under a third (30.2%) resided in cities or towns and a quarter (24.5%) resided in rural settlements.

Majority of respondents were never married (73.8%). For education, over a third (35.5%) had primary education or less; over a quarter (27.2%) had junior secondary education while just under a fifth had senior secondary school (17.3%) and tertiary education and over (19.9%). Close to two fifths (37.5%) of respondents were not employed; while over one quarter were employed in either the public (10.5%) or private sector (15.7%). Just over one in every ten (11.2%) were self-employed, while close to a fifth (18.8%) were either home makers or students; while under a tenth (6.4%) were retired from work.

Prevalence of NCD risk factors in the sampled population was as follows; smoking, 11.6%; alcohol consumption, 17.3%; poor physical activity, 48.9%; poor fruit and vegetable consumption, 82.5%; and overweight/obesity, 41.3%.

## Prevalence of multimorbidity

Prevalence of multimorbidity in the sampled population was estimated at 5.4%, while 24.2% of the sampled people reported a single NCD condition (Table 2). The prevalence of multimorbidity was high among females (6.6%), formerly married (20.4%), and increased with age, with 18% of individuals aged 65 years and above reporting multimorbidity. As regards education, people with primary or less education (9.5%) had the highest multimorbidity prevalence, while for wealth status people in the lowest quintiles (6.4%) had the highest prevalence. Among the behavioural risk factors, only alcohol consumption (6.4%) was significantly associated with multimorbidity. Similarly, 5.6% among people who had multiple NCD risk factors reported multimorbidity.

## Sociodemographic and behavioural correlates of multimorbidity

Sex was a significant correlate of multimorbidity when using behavioural and socioeconomic factors as covariates, with women observed to be 3 times (AOR = 3.34, 95% C.I. = 1.22–21.3) more likely to report multimorbidity than men (Table 3). It was also observed that age was a significant correlate of multimorbidity. For example, young people aged below 24 years were less likely to report multimorbidity (AOR = 0.01, 95% C.I. = 0.00–0.07) compared to elderly people aged 65 years and above. Currently married people were less likely to report multimorbidity compared to formerly married people when controlling for individual and behavioural risk factors (AOR = 0.24, 95% C.I. = 0.07–0.80). For wealth status of individuals in the 2nd wealth quintile (AOR = 0.19, 95% C.I. = 0.03–0.73) were found to be less likely to report multimorbidity than those in the 5th quintile, when adjusting for behavioural risk factors and socioeconomic covariates. For behavioural risk factors, alcohol consumption (AOR = 4.80, 95% C.I. = 1.16–19.8) and overweight/obesity (AOR = 1.44, 95% CI = 1.12–2.61) were significantly associated with multimorbidity, with alcohol consumers and people who were overweight/obese showing higher odds of multimorbidity.

**Table 2. Prevalence of number of chronic NCD conditions by socioeconomic and behavioural characteristics of the study population.**

| Variable | 0 chronic condition (n = 829) | 1 chronic condition (n = 285) | ->=2 chronic conditions (n = 63) |
|---|---|---|---|
| | % | % | % |
| **Sex***** | | | |
| Male | 81.9 | 15.7 | 2.5 |
| Female | 65.3 | 28.0 | 6.6 |
| **Age***** | | | |
| <=24 | 85.6 | 13.7 | 0.7 |
| 25–34 | 83.8 | 14.9 | 1.3 |
| 35–44 | 71.9 | 24.5 | 3.6 |
| 45–54 | 60.8 | 32.3 | 6.9 |
| 55–64 | 42.7 | 38.7 | 18.7 |
| 65+ | 30.0 | 52.0 | 18.0 |
| **Marital status***** | | | |
| Never-married | 77.1 | 19.4 | 3.5 |
| Currently-married | 56.3 | 38.2 | 5.5 |
| Formerly-married | 42.6 | 37.0 | 20.4 |
| **Education***** | | | |
| Primary or less | 55.6 | 34.9 | 9.5 |
| Secondary | 79.4 | 18.3 | 2.3 |
| Tertiary or higher | 77.8 | 18.3 | 3.9 |
| **Residence***** | | | |
| Cities and towns | 77.2 | 19.4 | 3.4 |
| Urban villages | 68.4 | 24.9 | 6.7 |
| Rural villages | 66.0 | 28.8 | 5.2 |
| **Work status***** | | | |
| Public sector | 63.1 | 30.3 | 6.6 |
| Private sector | 78.6 | 19.8 | 1.6 |
| self-employed | 69.2 | 26.2 | 4.6 |
| Not employed | 67.7 | 24.5 | 7.8 |
| Home-maker/student | 77.5 | 20.6 | 1.8 |
| Retired/other | 59.5 | 31.1 | 9.5 |
| **Wealth status** | | | |
| Lowest | 69.7 | 23.9 | 6.4 |
| Second | 68.4 | 26.6 | 5.1 |
| Middle | 71.5 | 23.0 | 5.5 |
| Fourth | 69.6 | 26.6 | 3.8 |
| Highest | 73.2 | 20.9 | 6.0 |
| **Smoking** | | | |
| Yes | 70.3 | 24.0 | 5.7 |
| No | 71.3 | 25.7 | 2.9 |
| **Poor physical activity** | | | |
| Yes | 73.4 | 22.6 | 4.0 |
| No | 69.3 | 24.8 | 6.0 |
| **Poor fruit/vegetable consumption** | | | |
| Yes | 70.4 | 24.3 | 5.2 |
| No | 70.2 | 23.1 | 6.6 |
| **Alcohol consumption***** | | | |
| Yes | 70.0 | 23.6 | 6.4 |

(*Continued*)

**Table 2.** (Continued)

|  | 0 chronic condition (n = 829) | 1 chronic condition (n = 285) | ->=2 chronic conditions (n = 63) |
|---|---|---|---|
| No | 79.9 | 18.6 | 1.5 |
| **Overweight/obesity**\*\*\* |  |  |  |
| Yes | 61.3 | 30.5 | 8.2 |
| No | 77.5 | 19.4 | 3.1 |
| **Overall** | **70.4** | **24.2** | **5.4** |

\*\*\*statistically significant at 5% level.

## Discussion

Multimorbidity is an emerging problem in LMICs and requires a holistic approach of health care system deliveries. Prevalence of multimorbidity in the sampled population for this study was 5.4%. Botswana is still at the early stages of nutritional and epidemiological transition. As a result it is expected that the clustering of NCDs in the population is relatively low. However, a higher prevalence of NCD multimorbidity has been observed in the older population aged 50 + years in this study and in other studies across several countries of the world [14–16]. It has been observed that in SAGE countries such as Mexico (64%) South Africa (63%), Russia (72%) and European countries such as Finland (68%), Poland (69%) and Spain (69%) [17], the prevalence of multimorbidity is high among the older population aged 50 years and above. In the African region, estimates on the prevalence of multimorbidity are still not available for many countries and the studies are limited. Due to rapid changes in unplanned urbanization, ageing trends, unhealthy dietary patterns, sedentary lifestyles, tobacco and alcohol use, multimorbidity is poised to increase substantially in the coming decades in Africa [18]. Meanwhile, the observed variations in multimorbidity between countries may be explained by several factors including; socio-economic development differences and different age compositions of study populations [19].

Gender differences were observed in the prevalence of multimorbidity. Women were found to have higher odds of reporting multimorbidity than men. This finding corroborates several other studies which have shown an increased prevalence of multimorbidity among women than men [20–23]. Other studies indicate that the clustering of NCD risk factors among women [24] explains in part, the consequent clustering of NCDs among women. Similarly, gender differences in multimorbidity observed in this study may be explained by high prevalence of risk factors such as poor physical activity and overweight/obesity which were found to be high among women.

Multimorbidity increased with age, and was more prevalent among the poor. Similarly, other studies also found that increasing age, and low wealth status were significant correlates of reporting multiple NCD conditions [14,16,21,22]. Most studies, in both developed and developing countries consistently identify aging as a significant correlate of multimorbidity [21,22]. This is mainly because NCD conditions converge in old age [23,25,26]. On the other hand, multinomial logistic regression analysis results in this study did not show any significant association between multimorbidity and place of residence, marital status, and education level. This indicates that there are no marital, educational and residential differences in multimorbidity. Thus, multimorbidity in the sampled population cuts across various marital, residential and educational groups.

Among the behavioural risk factors, alcohol consumption and overweight/obesity were significantly associated with multimorbidity. This association showed that people who reported

**Table 3. Results of multinomial logistic regression analysis for the outcome-the presence of multiple (two or more) chronic NCDs over no NCD condition.**

| Factors | Model I-Single NCD Condition/no NCD condition | Model II- Multimorbidity/no NCD condition | Model III-Multimorbidity/no NCD condition |
|---|---|---|---|
| | AOR C.I. | AOR C.I. | AOR C.I. |
| **Sex** | | | |
| Male | 1.00 | 1.00 | 1.00 |
| Female | 0.43*** (0.28–0.65) | 1.49*** (1.17–1.94) | 3.34*** (1.22–21.3) |
| **Age** | | | |
| ≤24 | 0.07*** (0.03–0.24) | 0.01*** (0.00–0.07) | 0.01*** (0.00–0.07) |
| 25–34 | 0.10*** (0.04–0.27) | 0.02*** (0.00–0.10) | 0.02*** (0.00–0.12) |
| 35–44 | 0.20*** (0.09–0.49) | 0.08*** (0.02–0.34) | 0.09*** (0.02–0.37) |
| 45–54 | 0.25*** (0.12–0.62) | 0.17*** (0.05–0.64) | 0.19*** (0.05–0.70) |
| 55–64 | 0.44 (0.18–1.05) | 0.68 (0.21–2.18) | 0.68 (0.21–2.18) |
| 65+ | 1.00 | 1.00 | 1.00 |
| **Marital status** | | | |
| Never-married | 1.03 (0.50–2.15) | 0.76 (0.28–2.04) | 0.72 (0.27–1.91) |
| Currently-married | 1.20 (0.57–2.53) | 0.24*** (0.07–0.80) | 0.23*** (0.06–0.76) |
| Formerly-married | 1.00 | 1.00 | 1.00 |
| **Education** | | | |
| Primary or less | 1.00 | 1.00 | 1.00 |
| Secondary | 1.39 (0.77–2.51) | 0.73 (0.21–2.56) | 0.82 (0.23–2.86) |
| Tertiary or higher | 1.10 (0.67–1.81) | 0.72 (0.23–2.25) | 0.79 (0.25–2.48) |
| **Residence** | | | |
| Cities and towns | 1.00 | 1.00 | 1.00 |
| Urban villages | 0.95 (0.58–1.56) | 1.09 (0.36–3.28) | 1.14 (0.38–3.43) |
| Rural villages | 1.11 (0.72–1.72) | 1.46 (0.62–3.45) | 1.50 (0.63–3.52) |
| **Work status** | | | |
| Public sector | 1.28 (0.58–2.83) | 0.89 (0.24–3.33) | 0.91 (0.24–3.37) |
| Private sector | 1.08 (0.50–2.36) | 0.53 (0.10–2.70) | 0.49 (0.09–2.48) |
| self-employed | 1.13 (0.51–2.50) | 0.86 (0.20–3.60) | 0.80 (0.19–3.32) |
| Not employed | 0.80 (0.40–1.61) | 0.96 (0.31–2.99) | 0.94 (0.30–2.92) |
| Home-maker/student | 1.19 (0.54–2.61) | 0.84 (0.18–3.93) | 0.83 (0.18–3.87) |
| Retired/other | 1.00 | 1.00 | 1.00 |
| **Wealth status** | | | |
| Lowest | 0.57 (0.30–1.10) | 0.32 (0.09–1.16) | 0.31 (0.08–1.12) |
| Second | 0.71 (0.40–1.28) | 0.20*** (0.05–0.75) | 0.19*** (0.05–0.73) |
| Middle | 0.71 (0.40–1.26) | 0.33 (0.10–1.10) | 0.32 (0.10–1.07) |
| Fourth | 1.16 (0.70–1.90) | 0.38 (0.11–1.22) | 0.38 (0.12–1.20) |
| Highest | 1.00 | 1.00 | 1.00 |
| **Smoking** | | | |
| Yes | | | 0.05 (0.00–0.43) |
| No | | | 1.00 |
| **Alcohol consumption** | | | |
| Yes | | | 4.80*** (1.16–19.8) |
| No | | | 1.00 |
| **Poor fruit/Vegetable consumption** | | | |
| Yes | | | 0.41 (0.07–2.22) |
| No | | | 1.00 |
| **Poor Physical activity** | | | |

(*Continued*)

**Table 3.**  (Continued)

| Factors | Model I-Single NCD Condition/no NCD condition | Model II- Multimorbidity/no NCD condition | Model III-Multimorbidity/no NCD condition |
|---|---|---|---|
| | AOR C.I. | AOR C.I. | AOR C.I. |
| Yes | | | 1.22 (0.32–4.63) |
| No | | | 1.00 |
| **Overweight/obesity** | | | |
| Yes | | | 1.44*** (1.12–2.61) |
| No | | | 1.00 |

\*\*\*statistically significant at 5% level; Model I- dependent variable = Single NCD condition/No NCD condition; this model includes socioeconomic variables as covariates. Model II- dependent variable = multimorbidity/No NCD condition; this model also includes socioeconomic variables as covariates; and Model III– dependent variable = multimorbidity/no NCD condition; this model includes NCD risk factors and socioeconomic variables as covariates.

alcohol consumption and those who were overweight/obese were more likely to report multimorbidity compared to those who did not. Some studies have shown similar findings where alcohol consumption and overweight/obesity were associated with high prevalence of multimorbidity [27]. The strong link between alcohol, overweight/obesity and multimorbidity in this study and in literature [28,29] indicates the need to support calls by WHO to implement evidence-based strategies to reduce harmful use of alcohol and to encourage healthy diets and physical activity. This is because alcohol consumption is a risk factor for overweight/obesity which in turn is a risk factor for several diseases and can exacerbate existing diseases particularly NCDs and can complicate the management of other chronic diseases [30,31]. Prevalence of overweight/obesity (41.3%) and consumption of alcohol was high (17.3%) in the sampled population and consistent with the previous finding, alcohol consumers in Botswana are generally hazardous drinkers and are often overweight/obese [6]. This possibly explains why alcohol consumers and overweight/obese individuals were more likely to have multimorbidity in this study.

The main strength of our study is that it is a large population-based study with a very high response rate. Data for this study contained information on potential confounding factors, with a low proportion of missing information making the study more comparable. Moreover, this is the first study on NCD multimorbidity among adults in Botswana; as a result it serves to provide the basis for further research. There are also limitations observed in this study. First, the study used a cross-sectional design, and it was not possible to establish the causal relationship between explanatory variables and multimorbidity. Furthermore, the sample for the NCDs study was not designed to be representative of the whole of Botswana. However, the findings of this study give an indication of emerging patterns of NCD multimorbidity and its associated factors in the country. The study used data that were collected from self-reports which may not necessarily be accurate, particularly social desirability effect. However we do not expect this limitation to have compromised the results to an extent that it renders them inadequate.

## Conclusion

This study set out to assess prevalence of multimorbidity and its correlates among adults in Botswana. The study found that NCD multimorbidity was more prevalent particularly among women, alcohol consumers and people who were overweight/obese. As the elderly population is likely to increase in the future both in proportion and in size, the prevalence of multimorbidity of NCDs is expected to go up. This will put greater pressure and challenges on the health

care system that should help the people with multiple chronic conditions. As a result, the health care system should adopt holistic approach of health care services to meet the needs of those suffering from multimorbidity.

## Acknowledgments

The authors wish to thank the University of Botswana for availing the time and resources to undertake this study. This paper is a part of the larger study on "Chronic Non communicable Diseases in Botswana: Chronic Disease Prevalence, Health Care Utilization, Health Expenditure and the Life course (NCD study)". We also wish to thank the study participants.

## Author Contributions

**Conceptualization:** Mpho Keetile, Kannan Navaneetham, Gobopamang Letamo.

**Formal analysis:** Mpho Keetile.

**Writing – original draft:** Mpho Keetile.

**Writing – review & editing:** Kannan Navaneetham, Gobopamang Letamo.

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
