## [Decision Letter · Decision Letter 0]

7 May 2020

PONE-D-20-04617

Prevalence and correlates of multimorbidity among adults in Botswana: A cross-sectional study

PLOS ONE

Dear Dr Keetile,

Thank you for submitting your manuscript to PLOS ONE. After careful consideration, we feel that it has merit but does not fully meet PLOS ONE’s publication criteria as it currently stands. Therefore, we invite you to submit a revised version of the manuscript that addresses the points raised during the review process.

We would appreciate receiving your revised manuscript by Jun 21 2020 11:59PM. To enhance the reproducibility of your results, we recommend that if applicable you deposit your laboratory protocols in protocols.io, where a protocol can be assigned its own identifier (DOI) such that it can be cited independently in the future. For instructions see: http://journals.plos.org/plosone/s/submission-guidelines#loc-laboratory-protocols

We look forward to receiving your revised manuscript.

Kind regards,

William Joe

Academic Editor

PLOS ONE

Journal Requirements:

2. Please ensure you have thoroughly discussed any potential limitations of this study within the Discussion section.

3. In the ethics statement in the Methods and online submission information, please ensure that you have specified how verbal consent was documented and witnessed.

Additional Editor Comments (if provided):

Reviewers' comments:

Reviewer's Responses to Questions

**Comments to the Author**

1. Is the manuscript technically sound, and do the data support the conclusions?

Reviewer #1: Partly

Reviewer #2: Yes

2. Has the statistical analysis been performed appropriately and rigorously? 

Reviewer #1: I Don't Know

Reviewer #2: I Don't Know

3. Have the authors made all data underlying the findings in their manuscript fully available?

Reviewer #1: Yes

Reviewer #2: Yes

4. Is the manuscript presented in an intelligible fashion and written in standard English?

Reviewer #1: No

Reviewer #2: Yes

5. Review Comments to the Author

Reviewer #1: The authors present a cross-sectional study of 1178 persons aged >=15 years examining the socioeconomic and behavioural factors associated with multimorbidity in Botswana. They find women, those consuming excessive alcohol and are overweight/obese to be at higher odds of reporting being multimorbid.

1. Abstract & Introduction: Multimorbidity and adjustment covariates could be defined in the abstract. The authors have not provided Botswana specific estimates with regards to burden of non-communicable disease. The study fails to address findings from previous literature to make a stronger case for their research question.

2. Methods: The authors could mention the inclusion and exclusion criteria for population. This section could be more concise.

3. Statistical analysis: The text seems unclear on what is the dependent variable. The authors should specify the tests used to create table 1 and 2. There is no clear mention if statistical tests to assess the fit of logistic model were assessed. Adjustments covariate. The authors should clarify what they mean by “Δcomplex sample module from SPSS was adoptedΔ”

4. Tables & Results: The authors use Multinomial logistic regression analysis for table 3 analysis. The write-up of results including table 3 content appear more to be multivariable logistic regression rather than multinomial logistic regression. In table 3, the multiple categories of age, work status and wealth status could be collapsed. Model 2 of table 3 is adjusted for Multiple NCD risk factors score, which created using of daily smoking, consuming less than 5 servings of fruit and vegetables per day, low physical activity, high body mass index (=>25), and self-reported diagnosis of hypertension. The model 2 is also then adjusted for some of the individual variables. The footnotes of all tables could be more elaborative.

5. Discussion: The language of discussion appear to be more like the results section. Strengths and limitations of the study have not been mentioned. This section could be rewritten taking into consideration recent publications.

6. References: In some places, references have not been added even though they have been mentioned. For example: page 4: “ΔThese were mainly based on the WHO Training Manual on the Study on Global Ageing and Adult Health (SAGE) and other documents were used for reinforcementΔ”.

7. Writing Quality: The authors could revise the manuscript for readability, as in the current version there are typos and grammatical mistakes.

Reviewer #2: a) In introduction in line 11-12 you have mentioned about the increasing burden of NCDs, it would be better if you can explain with some sought of existing data available on this issue.

b) In introduction, explain the reasons behind NCDs in little explicit manner, such as how rapid unplanned urbanization is linked to NCDs. (so that it would be easy for reader to have a knowledge regarding that issue)

c) Reason behind taking wealth status not wealth index.

d) Part of the article where you are explaining socio demographic and behavioral correlates of multi morbidity, you can mention the impact of residence and education as well.

e) Conclusion and recommendation can have more points in relation to the present change in Botswana and as a developing nation what actions can be taken.

6. PLOS authors have the option to publish the peer review history of their article (what does this mean?). If published, this will include your full peer review and any attached files.

Reviewer #1: No

Reviewer #2: No

---

## [Author Response · Author response to Decision Letter 0]

22 Jul 2020

Response to Reviewers’ Comments

PONE-D-20-04617

Prevalence and correlates of multimorbidity among adults in Botswana: A cross-sectional study

PLOS ONE

Comments to the Author

Reviewer #1

 Comment

 The authors present a cross-sectional study of 1178 persons aged >=15 years examining the socioeconomic and behavioural factors associated with multimorbidity in Botswana. They find women, those consuming excessive alcohol and are overweight/obese to be at higher odds of reporting being multimorbid.

Response

Thank you for the comment.

Comment

Abstract & Introduction: Multimorbidity and adjustment covariates could be defined in the abstract. The authors have not provided Botswana specific estimates with regards to burden of non-communicable disease. The study fails to address findings from previous literature to make a stronger case for their research question.

Response

We have mentioned multimorbidity and adjustment of covariates as suggested by the reviewer. We have also done more literature review and provided background information on non-communicable diseases and their risk factors in Botswana as suggested by the reviewer.

Comment

Methods: The authors could mention the inclusion and exclusion criteria for population. This section could be more concise.

Response

We have indicated in the method section, the inclusion and exclusion criteria for sampled population. We have made efforts to make the section more concise. 

Comment

Statistical analysis: The text seems unclear on what is the dependent variable. 

Response

The dependent/outcome variable for this study is multimorbidity. We have provided an explicit explanation of how we measured this outcome under the measures subsection.

Comment

The authors should specify the tests used to create table 1 and 2. There is no clear mention if statistical tests to assess the fit of logistic model were assessed. Adjustments covariate. The authors should clarify what they mean by “Δcomplex sample module from SPSS was adoptedΔ”

Response

 We have indicated that percentages were used in table 1 to indicate the distribution of the sampled population and chi-square tests (table2) were used to assess the association between multimorbidity and independent variables. The goodness of fit of different models were assessed using the likelihood ratio test and model pseudo R square value. We have indicated that the complex sample command in SPSS is used to account for the multiple stages of sampling, when analyzing data derived using multistage sampling design.

Comment

Tables & Results: The authors use Multinomial logistic regression analysis for table 3 analysis. The write-up of results including table 3 content appear more to be multivariable logistic regression rather than multinomial logistic regression. 

Response

Thank you for the comment. Kindly note that we had put an abridged table showing logistic regression results for multimorbidity and not single morbidity. We have since put the single morbidity column to indicate that multinomial logistic regression analysis was done. Also, the study is more interested to look at the covariates for outcome of more than two NCD conditions relative to no NCD condition. 

Comment

In table 3, the multiple categories of age, work status and wealth status could be collapsed. 

Response

Thank you for the comment; although we agree with the possibility of collapsing the above categories, we thought that collapsing them further would pelt the in-group dynamics and variations. Moreover, we were guided by previous literature in coding these variables.

Comment

Model 2 of table 3 is adjusted for Multiple NCD risk factors score, which created using of daily smoking, consuming less than 5 servings of fruit and vegetables per day, low physical activity, high body mass index (=>25), and self-reported diagnosis of hypertension. The model 2 is also then adjusted for some of the individual variables. The footnotes of all tables could be more elaborative.

Response

Thank you. We have rearranged this section. We have also provided more elaborate explanation in the footnotes.

 

Comment

Discussion: The language of discussion appears to be more like the results section. Strengths and limitations of the study have not been mentioned. This section could be rewritten taking into consideration recent publications.

Response

We have re-looked into this section and added recent publications. We have also added the strengths and limitations sections

Comment

References: In some places, references have not been added even though they have been mentioned. For example: page 4: “ΔThese were mainly based on the WHO Training Manual on the Study on Global Ageing and Adult Health (SAGE) and other documents were used for reinforcementΔ”.

Comment

Writing Quality: The authors could revise the manuscript for readability, as in the current version there are typos and grammatical mistakes.

Response: 

The manuscript has been revised and thoroughly checked the typos and grammatical mistakes. 

Reviewer #2

Comment 

In introduction in line 11-12 you have mentioned about the increasing burden of NCDs, it would be better if you can explain with some sought of existing data available on this issue.

Response 

We have provided more data on the burden of NCDs. Furthermore, it has to be noted that this is an inaugural study to cover many other NCDs, besides hypertension, diabetes and cancer. There is paucity of literature

Comment

In introduction, explain the reasons behind NCDs in little explicit manner, such as how rapid unplanned urbanization is linked to NCDs. (so that it would be easy for reader to have a knowledge regarding that issue)

Response

We have provided more information and explained how rapid urbanization and nutrition transition has led to changes in disease patterns in Botswana 

Comment

Reason behind taking wealth status not wealth index.

Response

Thanks for the comment. Wealth status is derived from the wealth index constructed through asset indicators. Of course wealth index is proxy indicator of wealth status which generally measure the i socioeconomic status of individuals.

Comment

Part of the article where you are explaining socio demographic and behavioural correlates of multi morbidity; you can mention the impact of residence and education as well.

Response

We indicate in the discussion section that education and residence were not significant covariates of multimorbidity, indicating that there are no educational and residential differences in multimorbidity. Thus, multimorbidity in the sampled population cuts across residential and educational divides.

Comment

Conclusion and recommendation can have more points in relation to the present change in Botswana and as a developing nation what actions can be taken.

Response

The conclusion and recommendation section has been modified which includes policy action needed to respond to the increase in the older population in the future in Botswana.

---

## [Decision Letter · Decision Letter 1]

4 Sep 2020

Prevalence and correlates of multimorbidity among adults in Botswana: A cross-sectional study

PONE-D-20-04617R1

Dear Dr. Keetile,

We’re pleased to inform you that your manuscript has been judged scientifically suitable for publication and will be formally accepted for publication once it meets all outstanding technical requirements.

Kind regards,

William Joe

Academic Editor

PLOS ONE

Additional Editor Comments (optional):

Reviewers' comments:

Reviewer's Responses to Questions

**Comments to the Author**

1. If the authors have adequately addressed your comments raised in a previous round of review and you feel that this manuscript is now acceptable for publication, you may indicate that here to bypass the “Comments to the Author” section, enter your conflict of interest statement in the “Confidential to Editor” section, and submit your "Accept" recommendation.

Reviewer #1: (No Response)

2. Is the manuscript technically sound, and do the data support the conclusions?

Reviewer #1: Partly

3. Has the statistical analysis been performed appropriately and rigorously? 

Reviewer #1: (No Response)

4. Have the authors made all data underlying the findings in their manuscript fully available?

Reviewer #1: Yes

5. Is the manuscript presented in an intelligible fashion and written in standard English?

Reviewer #1: No

6. Review Comments to the Author

Reviewer #1: (No Response)

7. PLOS authors have the option to publish the peer review history of their article (what does this mean?). If published, this will include your full peer review and any attached files.

Reviewer #1: No

---

## [Editor Report · Acceptance letter]

10 Sep 2020

PONE-D-20-04617R1 

Prevalence and correlates of multimorbidity among adults in Botswana: A cross-sectional study 

Dear Dr. Keetile:

I'm pleased to inform you that your manuscript has been deemed suitable for publication in PLOS ONE. Congratulations! Your manuscript is now with our production department. 

Kind regards, 

on behalf of

Dr. William Joe 

Academic Editor

PLOS ONE